# LITA: LIGHT AGENT UNCOVERS THE AGENTIC CODING CAPABILITIES OF LLMS

## ABSTRACT

Large language models (LLMs) are increasingly being applied to programming tasks, ranging from single-turn code completion to autonomous agents. Current code agent designs frequently depend on complex, hand-crafted workflows and tool sets. However, this reliance on elaborate scaffolding presents several challenges: agent performance becomes overly dependent on prompt tuning and custom design choices, heavy human intervention obscures a model's true underlying capabilities, and intricate pipelines are costly to build and maintain. Furthermore, optimizing complex task prompts increases the risk of data leakage. Currently, when introducing new models, LLM providers like OpenAI and Anthropic often publish benchmark scores to demonstrate their models' coding proficiency, but keep their proprietary evaluation frameworks confidential. To address these limitations, we introduce *Lita* (**Lit**e **A**gent), which operationalizes *liteness*, a principle of minimizing manual design while retaining the essential elements of a fully autonomous agent. Lita enables a more faithful and unified evaluation without elaborate scaffolding. Experiments on the Aider Polyglot and SWE-Bench with frontier models demonstrate that Lita achieves competitive or superior performance compared to workflow-based and agentic baselines. Crucially, Lita also consumes fewer tokens and requires significantly less design effort. Our results suggest that Lita is sufficient to reveal the underlying coding competence of modern LLMs. Finally, we propose the **Agent Complexity Law**: *the performance gap between agents of varying complexity, from simple to sophisticated designs, will shrink as the core model improves, ultimately converging to a negligible difference.*

## 1 INTRODUCTION

Large language models (LLMs) have rapidly transformed the way people work, study, and conduct research. Beyond their role in natural language understanding, recent advances have demonstrated their capacity to assist in highly specialized domains, ranging from mathematics to scientific discovery (OpenAI, b; Wang et al., 2023). Among these domains, programming has emerged as one of the most impactful frontiers. Models from Anthropic, Gemini, OpenAI, DeepSeek, and Qwen have shown strong performance in software engineering tasks, while their recent releases consistently emphasize coding ability as a core benchmark of progress. For example, Claude Opus 4 attains 72.5% on SWE-Bench (Anthropic, 2025a), a benchmark that measures performance in real-world software engineering tasks (Jimenez et al., 2024). The increasing integration of LLMs into software development workflows has also been seen as a step toward artificial general intelligence (AGI), since coding requires precise reasoning, planning, and interaction with complex systems.

Within the broader field of LLMs for code, researchers and practitioners have explored a range of system designs to leverage these models effectively. Besides generating the solution in a single-turn completion usually seen in simple tasks like HumanEval (Chen et al., 2021a), current designs can be categorized into two paradigms. *First*, to incorporate richer feedback, the **workflow paradigm** including Agentless (Xia et al., 2024) and Aider (Aider-AI), introduces predefined, human-designed procedures, allowing the model to iteratively refine solutions within controlled steps. *Second*, more recently, the **agentic paradigm** has gained traction, where fully autonomous agents interact with external environments, execute code, and adjust their responses through trial and error. Early systems such as SWE-Agent (Yang et al., 2024) pioneered this approach by parsing model outputs into rule-

based tool invocations, while newer frameworks such as OpenHands (Wang et al., 2025) leverage function calling abilities to streamline interaction between the model and coding environments.

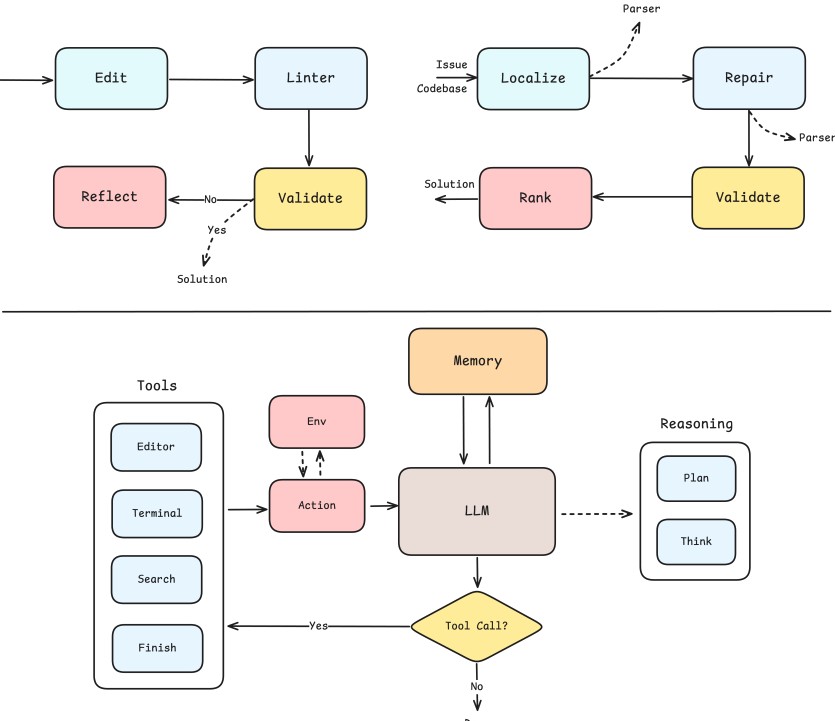

Figure 1: The upper left sub-diagram shows the workflow agent for Aider's polyglotAider-AI benchmark. The upper right sub-diagram shows the workflow for Agentless Xia et al. (2024) testing of SWE-bench. The lower sub-diagram represents our Lita autonomous agent framework, with key modules including LLM, Memory, Tools, Reasoning and Environment.

Despite these advances, current approaches to coding with LLMs still face several fundamental challenges, which hinder robust evaluation of models' true capabilities and recognition of their limitations.

**CHALLENGE 1: Fairness.** Many existing frameworks are tightly coupled with particular models, making fair comparison difficult. Prompts and tools are often optimized for specific architectures. For example, both CodeX and OpenHands prompts are particularly well suited to GPT-series models, creating hidden advantages (OpenAI, a; OpenHands, a).

In workflow-based systems, such as Agentless, some stages are especially prone to failure depending on the underlying model. A weaker model may struggle with tasks like autonomous bug localization or program repair, but predefined workflows can mask its weakness by constraining the space of possible errors (e.g. GPT-4o compared with Claude-3.5-Sonnet, see Xia et al. (2024)).

Even on the same dataset, discrepancies in prompts, toolkits, or scoring protocols across companies lead to a misalignment of different models' evaluation (Gao et al., 2024; Zhuo et al., 2024a; OpenHands, b).

**CHALLENGE 2: Truthfulness.** Workflows introduce extensive human guidance, making it difficult to assess the intrinsic capabilities of models. This gap leads to inflated benchmark scores that may not translate to practical performance in real use (Liu et al., 2023; Mozannar et al., 2024).

Agent systems can still fall into similar traps: many benchmarks introduce over-engineered tool sets tailored to specific tasks, effectively "teaching to the test". Tool descriptions in some frameworks often encode workflow-like instructions, which implicitly steer models toward particular solutions (e.g. OpenHands' prompts tell how to solve SWE-Bench problems (OpenHands, b)). Such practices

risk undermining core abilities like autonomous planning and memory management, precisely the capabilities that stronger models ought to demonstrate.

**CHALLENGE 3: Overhead.** The complexity of heavily engineered workflows imposes high costs on both developers and users. Each new benchmark often requires significant prompt and tool re-tuning, thus may lead to poor portability across tasks. This distracts them from improving the intrinsic model capabilities and designing environments that genuinely test agent autonomy.

Elaborate workflows used for raising benchmark scores also introduce overhead on the user's end: increased token usage, longer interaction traces, and more complex context management. Ultimately, these bills will be paid by the users.

Based on these challenges, we argue that simplified agent designs can better expose the strengths and weaknesses of LLMs for coding, while reducing opportunities for hidden biases or benchmark-specific optimizations. By minimizing scaffolding, such designs maximize the space for autonomous exploration and provide a more faithful evaluation of model competence, especially when recent work has highlighted that evaluation is as critical as model development itself (Wei et al., 2025; Yao). Therefore, in this paper, we present Lita (Lite Agent), a lightweight agentic framework for evaluating and extending LLMs in coding tasks. Our key contributions are as follows.

- We introduce the concept of Lite Agent and implement a prototype system, Lita, which offers more authentic evaluation and strong adaptability across tasks and datasets.

- We propose a method for converting widely used coding benchmarks into multi-turn, agentic settings, enabling agents to autonomously complete tasks in a unified format.

- We empirically demonstrate the feasibility of Lita, comparing it against existing frameworks and conduct ablations to identify how minimal the design can be while still supporting effective performance.

- We propose the **Agent Complexity Law**: the performance gap between agents of varying complexity, from simple to sophisticated designs, will shrink as the core model improves, ultimately converging to a negligible difference.

We also conducted an extensive survey of prior work on LLMs for code, agent design philosophies, and corresponding benchmarks; for readability, we present this discussion in Appendix A.1. It is also worth noting that simplifying design for evaluation does not contradict practical prompt engineering. While minimal scaffolding is essential for revealing a model's intrinsic capabilities, in real-world applications prompt engineering remains valuable to maximize user experience.

## 2 METHOD

### 2.1 PRINCIPLES OF LITE AGENT

- *Decoupling the agent from specific LLMs and Tasks*
- *Simplicity over complexity*
- *Workflow-free, prioritizing autonomy*
- *Minimize prompt engineering; trust and harness the evolving capabilities of models*

— Lita Design Philosophy

To address the challenges identified above, we introduce the concept of Lite Agent. Our principle is that an LLM-based coding system should minimize manual scaffolding by keeping three core dimensions, i.e. the underlying LLM, the agent framework, and the environment (e.g. benchmarks), as decoupled as possible. To this end, we have summarized the four philosophies of lita design.

An agent system typically consists of three elements: the LLM, tools, and the environment. It calls tools to execute critical procedures for a task. In Lita, these will be invoked through function calls, which most modern models now support for autonomous interaction.

At the same time, a lite agent is designed to be minimal. It should contain only those tools strictly necessary to complete software engineering (SWE) tasks, while avoiding over-engineered or redundant toolkits. For Lita, its tool schema (the descriptions and argument specifications for each tool) should be compact and unambiguous, reducing opportunities for benchmark overfitting. This design philosophy ensures that agent performance reflects the model's own reasoning and decision-making ability, rather than intentional human optimizations.

## 2.2 COMPONENT DESIGN OF LITA

Except the environment, Lita consists of three key components: tools, reasoning, and memory, similar to Claude (Anthropic, 2025a). To collect information from environment, we first define a small set of tools:

- **Editor** - for creating, viewing or modifying files
- **Terminal** - for executing commands or running tests
- **Search** - for searching a code snippet in files under a directory
- **Finish** - for signaling task completion

The reasoning module is designed to support structured thinking, we implement **Think** and **Plan** tools to interact with this module. These allow the model to record self-reflection or outline next steps explicitly, without embedding workflow-like instructions into the environment.

The memory module manages context, for which we implement two strategies:

- Linear memory - accumulating the entire interaction history
- Summarized memory - letting the LLM decide when to condense parts of the history into shorter summaries, which can be invoked through **Summary** tool calls

To best reveal a model's capacity for long-context management, we adopt linear memory by default, with summarized memory provided as an option.

Tool schemas are human-designed only to the extent of clarifying function semantics and parameters. We then let the LLM itself refine the wording only to ensure they are easy to understand for the model, avoiding the pitfalls of heavy prompt engineering.

Our survey of existing code agents suggests that these components are sufficient to cover typical SWE tasks such as editing, terminal interaction, and testing. Moreover, we verify that tools in other agent systems can be decomposed into them. For their necessity, we will conduct ablation studies in Section 3.3. Features such as retrieval or web search are left for future research.

## 2.3 BENCHMARK TRANSFORMATION

One of our contributions is the transformation of widely used code benchmarks into agentic form. Our goal is to enable multi-turn, autonomous evaluation while adhering to two design principles: (i) prompts and interactions should remain simple, avoiding model-specific optimizations; and (ii) agents should be evaluated in a unified format, ensuring fairness and portability.

Each benchmark instance is first reformulated into an initial user prompt with four following parts (see Figure 2). **Initial state** specifies the working directory and available files. **Task description** describes the objective of task instance, such as fixing a bug or completing a function. In some cases, detailed task statement may be provided in an external file for the agent to read. **Output state** indicates the final expected directory structure and which files contain the solution. **Validation steps** describe how the agent could verify its solution, such as executing commands or generating unit tests.

We apply this template to three frequently-used code benchmarks - HumanEval, Aider's Polyglot, and SWE-Bench Verified, harmonizing their file structures and task descriptions so that agents can be assessed under the same evaluation protocol. This conversion allows benchmarks to shift from static completion or editing tasks into dynamic, interactive environments.

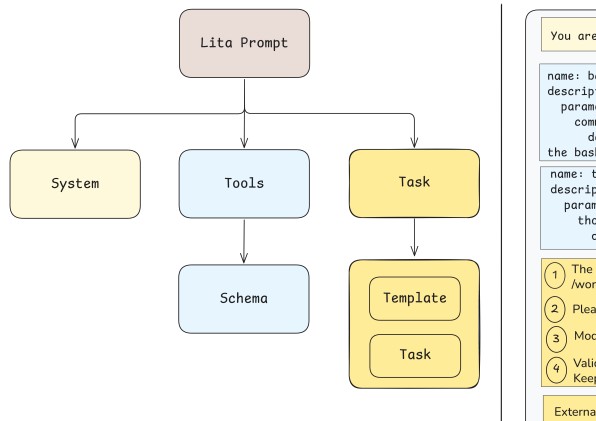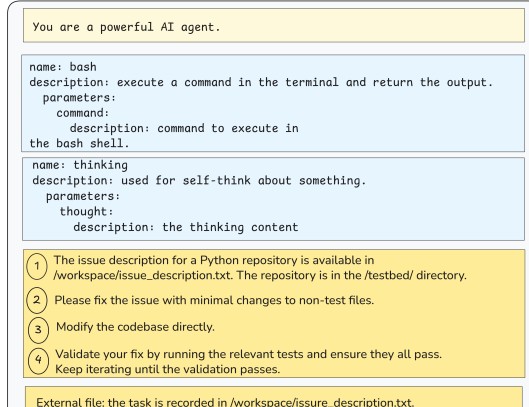

Figure 2: This figure presents an agent's prompt design. The left diagram shows the general components of an agent system prompt, while the right provides a specific example of Lita on SWE-bench. Specifically, the task template requires four essential components: Initial State, Task Description, Output State, and Validation Steps.

## 2.4 MEASURING LITENESS

Finally, we propose a quantitative measure of *liteness* to capture the complexity of an agent design, which is called **Agent Intrinsic Complexity**. This measure considers two factors:

- Action count - the number of supported tools
- System preloaded tokens - the token cost of system-level content, including the system prompt, the initial user prompt (Section 2.3), and the tool schema

By combining these metrics, we provide a principled way to assess how lightweight a given agent framework is. This allows us to systematically compare Lita with existing workflow-heavy and agent-rich baselines, and to analyze how design complexity impacts both fair and truthful evaluation and model performance.

## 3 EXPERIMENT AND RESULT

We evaluate Lita across a range of benchmarks, models, and scaffolding paradigms. For better comparison, we implement two editing formats - one based on git-diff blocks and the other on string replacement, illustrated in Appendix Figure 4. The string-replace version serves as the default implementation of Lita, while the diff-based variant is denoted as Lita-diff. We also include a **Terminal**-only variant, Lita-mini, to study the minimal agent design.

**Datasets**. Following Section 2.3, we convert three widely used coding benchmarks into our unified agentic format: HumanEval (function-level completion, low difficulty; results shown in Appendix Table 4), Aider's Polyglot (multi-programming language code generation, intermediate difficulty), and SWE-Bench Verified (real-world bug fixing, high difficulty).

**Models**. Our experiments span both proprietary and open-source models, covering a spectrum of capability. We include models from GPT and Claude families as well as the Qwen series, allowing us to examine performance trends from weaker to stronger models.

**Scaffoldings**. We compare:

- Workflow systems - Aider on Polyglot. On SWE-Bench, workflow baselines are omitted since recent evaluations only focus on agentic setups, which have already matched the performance of workflows.
- Agentic systems - OpenHands, open-sourced, as well as broadly validated in both academia and industry. To ensure fairness, we keep its system prompt but replace task prompts with

Table 1: Main results across models and scaffolds. Bold marks the best value within each LLM for the *In 50 Turns* budget and the lowest cost.

| LLM | Scaffold | Pass Rate (%) | | Edit (%) | Token Count (M) | | Cost ($) |
| --- | --- | --- | --- | --- | --- | --- | --- |
| | | In 2 Tests | In 50 Turns | | Input | Output | |
| Claude Opus 4 | Lita | 38.2 | **96.4** | 99.2 | 20.7 | 0.9 | **376.2** |
| | OpenHands | 52.3 | 95.4 | 98.8 | 34.2 | 1.0 | 587.9 |
| | Aider | 70.7 | - | 98.7 | - | - | - |
| Claude Sonnet 4 | Lita | 13.9 | **97.8** | 86.9 | 47.4 | 1.4 | **163.6** |
| | OpenHands | 15.1 | 96.0 | 96.8 | 66.7 | 1.5 | 222.0 |
| | Aider | 56.4 | - | 98.2 | - | - | - |
| Claude 3.7 Sonnet | Lita | 55.4 | **98.7** | 94.4 | 24.4 | 1.0 | **88.4** |
| | OpenHands | 56.3 | 98.2 | 90.6 | 51.1 | 1.9 | 181.5 |
| | Aider | 60.4 | - | 93.3 | - | - | - |
| GPT-5 | Lita | 85.1 | 96.0 | 98.5 | 7.0 | 0.7 | **15.4** |
| | OpenHands | 88.5 | **96.8** | 99.2 | 15.6 | 0.8 | 27.8 |
| | Aider | 86.0 | - | 88.0 | - | - | - |
| GPT-4.1 | Lita | 47.7 | **81.1** | 81.8 | 44.3 | 0.9 | **95.6** |
| | OpenHands | 43.2 | 81.1 | 88.5 | 69.7 | 0.8 | 145.7 |
| | Aider | 52.4 | - | 98.2 | - | - | - |
| GPT-4.1-mini | Lita | 25.9 | **67.8** | 74.8 | 100.7 | 1.3 | 42.4 |
| | OpenHands | 26.8 | 67.3 | 73.6 | 86.6 | 1.1 | **36.4** |
| | Aider | 32.4 | - | 92.4 | - | - | - |
| GPT-4o | Lita | 17.6 | **45.8** | 56.5 | 104.5 | 1.5 | **276.3** |
| | OpenHands | 15.2 | 41.2 | 66.4 | 108.3 | 1.2 | 283.2 |
| | Aider | 23.1 | - | 94.2 | - | - | - |
| GPT-4o-mini | Lita | 2.8 | **6.6** | 7.3 | 107.8 | 2.2 | **17.5** |
| | OpenHands | 2.3 | 4.2 | 31.0 | 149.3 | 1.5 | 23.3 |
| | Aider | 3.6 | - | 100.0 | - | - | - |

Lita's on Polyglot since its original ones are tightly coupled with SWE-Bench. We also include mini-SWE-agent, a rule-based terminal-only baseline, on SWE-Bench.

- Our framework (Lita) - lightweight design with minimal tools and action schema, with variants (Lita-diff, Lita-mini) for ablation and better comparison with both Aider and mini-SWE-agent.

**Metrics**. We measure task success rate (pass@1 against external test cases or resolution rate), token consumption, and per-tool call counts. On Polyglot, we additionally track the success rate of adhering to the diff-format edits as they reported on Aider's leaderboard. Its pass@1 after the first 2 unit tests and 50 interaction turns are both recorded. Detailed hyperparameter and runtime settings are provided in the Appendix A.4.

### 3.1 RESULTS ON AIDER'S POLYGLOT

Table 1 reports the results. We highlight three key observations:

(1) **Lita vs. OpenHands**. Across nearly all models, Lita achieves higher pass rates while consuming fewer tokens. This contrast suggests that OpenHands' heavy optimization for SWE-Bench has led to overfitting. When applied to Polyglot, a mid-difficulty benchmark, its performance lags behind Lita instead. Analysis of tool call logs (Appendix Figure 5 and Table 5) shows that Lita allocates more function calls to **Think** and **Plan**, indicating that its token budget is spent on reasoning rather than repetitive wasted edits.

(2) **Lita vs. Aider**. Workflow guidance in Aider reduces early-stage mistakes, yielding higher pass rates in the first two tests. However, agentic methods allow LLMs to autonomously gather information and recover from errors through trial and error. Stronger models can recover from early failure to achieve a higher score later and nearly solve Polyglot entirely. We argue that final

Table 2: Solved rate (%) on SWE-bench across models (rows) and agents (columns). The data in the official column is provided by the model providers.

| Model | Official | OpenHands | mini-SWE-agent | Lita | Lita-min |
|---|---|---|---|---|---|
| Qwen3-Coder 30B | 51.6 | 47.6 | - | - | - |
| Qwen3-Coder 480B | 67.0 | 64.6 | 55.4 | - | - |
| GPT-4.1-mini | 22.8 | 22.0 | 23.94 | 26.4 | 11.8 |
| GPT-4.1 | 52.0 | 48.6 | 39.58 | 35.6 | 19.6 |
| Claude 3.7 Sonnet | 61.0 | 58.0 | 52.8 | 53.0 | 48.6 |
| Claude Sonnet 4 | 69.4 | 68.0 | 64.8 | 62.0 | 57.8 |
| Claude Opus 4 | 69.2 | 67.8 | 67.6 | 62.6 | 55.2 |

resolution, rather than initial attempts, better reflects real-world usage, where agents may iterate until success.

(3) **File editing success rates**. For all frameworks, adherence to the diff format improves with model strength, reflecting better instruction-following ability. This trend reinforces that editing style interacts closely with model capability.

## 3.2 RESULTS ON SWE-BENCH VERIFIED

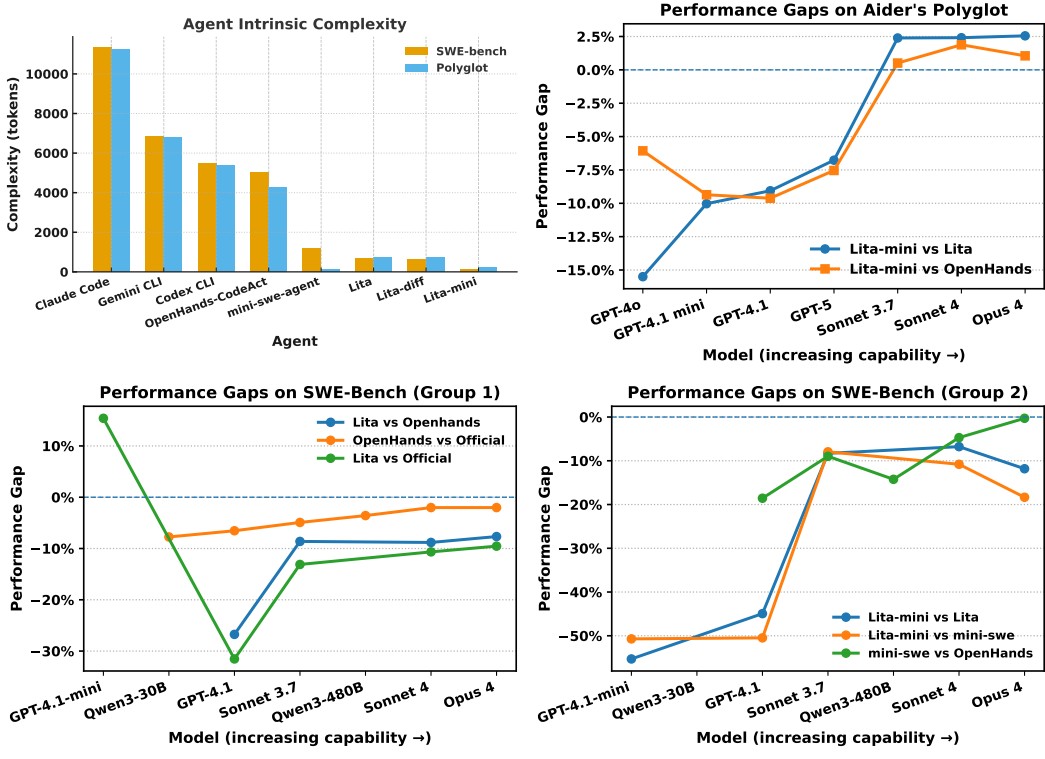

Figure 3: Agent Intrinsic Complexity and Performance gap between simple and complex agent.

Resolution rates are reported in Table 2, with cost statistics in the Appendix Table 6. To compare frameworks more systematically, we compute relative performance gaps between simple and complex designs (let $P$ to be performance, then $P_{\text{rel}} = (P_{\text{simple}} - P_{\text{complex}})/P_{\text{complex}}$), and plot these against model strength in Figure 3. We make three observations:

(1) **General vs. task-specific optimization**. Lita trails OpenHands slightly because Lita is a general framework, while OpenHands includes task-specific hints in its SWE-Bench prompts. Such hints risk data leakage and overfitting, as also reflected by OpenHands' weaker performance on Poly-

Table 3: Ablation study of different Lita variants across models.

| LLM | Variant | Pass rate in 2 tests | Pass rate in 50 turns | Edit (%) |
|---|---|---|---|---|
| GPT-4o | Lita-diff | 8.5 | 19.8 | 16.1 |
| | Lita-mini | 13.2 | 38.7 | - |
| | Lita | 17.6 | 45.8 | 56.5 |
| | Lita-reason | 11.2 | 41.4 | - |
| GPT-4.1-mini | Lita-diff | 25.8 | 62.9 | 63.2 |
| | Lita-mini | 31.8 | 61.0 | - |
| | Lita | 25.9 | 67.8 | 74.8 |
| | Lita-reason | 23.7 | 51.8 | - |
| GPT-4.1 | Lita-diff | 47.7 | 81.1 | 81.8 |
| | Lita-mini | 39.1 | 73.3 | - |
| | Lita | 48.1 | 80.6 | 86.5 |
| | Lita-reason | 31.7 | 68.8 | - |
| GPT-5 | Lita-diff | 85.7 | 94.9 | 77.4 |
| | Lita-mini | 34.2 | 89.5 | - |
| | Lita | 85.1 | 96.0 | 98.5 |
| | Lita-reason | 32.1 | 87.3 | - |
| Claude 3.7 Sonnet | Lita-diff | 51.9 | 97.2 | 87.0 |
| | Lita-mini | 55.4 | 98.7 | - |
| | Lita | 57.5 | 96.4 | 94.4 |
| | Lita-reason | 47.8 | 97.8 | - |
| Claude Sonnet 4 | Lita-diff | 13.9 | 97.8 | 86.9 |
| | Lita-mini | 17.3 | 97.8 | - |
| | Lita | 15.6 | 95.5 | 96.7 |
| | Lita-reason | 14.9 | 95.0 | - |
| Claude Opus 4 | Lita-diff | 34.7 | 94.2 | 88.2 |
| | Lita-mini | 38.2 | 96.4 | - |
| | Lita | 48.4 | 94.0 | 99.2 |

glot. Similarly, mini-SWE-agent, though minimal, embeds handcrafted rules that partially encode solutions.

(2) **Paradigm shift, not system error**. The performance gap between Lita and OpenHands remains within a reasonable range (around 10%) across all models, showing that differences are attributable to paradigm choice rather than implementation flaws in Lita. Moreover, the similar performance curves of Lita and OpenHands on Polyglot further support this interpretation.

(3) **Agent Complexity Law**. We observe a consistent trend: as model capability increases, the performance gap between frameworks of varying complexity shrinks. This holds across different baselines (OpenHands, mini-SWE-agent) as well as among Lita variants. On simpler tasks such as Polyglot, lightweight agents can even outperform more complex systems. Occasional outliers occur between adjacent models of the same generation, whose overall capabilities are similar but may fluctuate on specific tasks (e.g. Sonnet 4 vs Opus 4 in right-lower part of Figure 3). The initial dip of the curves in left-lower part of Figure 3 is due to outdated official results - both Lita and mini-SWE-agent surpass the reported GPT-4.1-mini baseline (OpenAI, 2025b).

These findings suggest that elaborate agent designs provide diminishing returns as model capacities scale. For robust evaluation, these designs may soon be unnecessary, allowing the intrinsic capabilities of models to be more clearly revealed and better guiding their future improvement.

## 3.3 ABLATION STUDIES

To assess which tools are necessary for Lita and how minimal an agent can be, we conduct ablations on Polyglot, which balances task difficulty with manageable evaluation cost. Besides the terminal-only Lita-mini and diff-based Lita-diff, we include Lita-reason with Terminal, Think and Plan tools for explicit reasoning.

Results are shown in Table 3. Two main patterns emerge:

(1) **File editing strategy matters**. Replacing diff-based editing with string replacement significantly improves edit success, especially for smaller models with weaker instruction-following ability. This aligns with observations in Section 3.1 and OpenAI's releases (OpenAI, 2025b).

(2) **Minimal tools suffice, but with trade-offs**. A terminal-only agent already achieves competitive results and can even surpass full Lita on strong models, which are better able to autonomously explore the way to edit files and interact with the production system. However, for weaker models, explicit editing and reasoning tools remain necessary. This demonstrates that Lita's chosen tool set in the default version provides a stable baseline for fair evaluation across model families.

## 4    DISCUSSION AND LIMITATIONS

Our study highlights both the promise and the limitations of lite agents in LLM-based coding systems.

**Failure cases**. While Lita generally outperforms existing agentic frameworks, we observe cases where even strong models (e.g. Claude 4) fail at the very first step of a task, whereas workflow-guided systems can still succeed. This suggests that minimal agents place greater demands on model robustness. When an initial plan is flawed, recovery depends on the model's capacity for self-correction.

**Self-exploration by stronger models**. In Lita, stronger models often start by exploring the directory structure before making edits. This exploratory strategy is particularly effective for complex projects and contrasts with Aider's fixed workflow, providing evidence that rigid workflows may constrain model autonomy.

**Liteness is not orthogonality**. A lightweight design does not imply that tools are interchangeable. For example, the Editor tool abstracts a set of frequently used coding actions; removing it cannot be compensated by other components without performance loss. Thus, "minimal" should be understood as "sufficient but not redundant", rather than "functionally non-overlapping".

**Limitations**. Our work also has several limitations. First, although we converted multiple benchmarks into agentic form, these datasets still represent a narrow slice of real-world software engineering. Future benchmarks could expand to multi-repository projects, collaborative development, or longer-term maintenance tasks. Second, since Lita is a prototype system, we didn't include advanced features such as retrieval, web search and multi-agent, or conduct post-training, which may be necessary for scaling to more complex tasks. Finally, while our evaluation focuses on fairness and truthfulness, we have not yet studied long-term human-agent interaction, which is essential for deployment in practical development environments.

## 5    CONCLUSION

This paper asked a simple but pressing question: *Is complex design really necessary for evaluating LLM-based coding agents?* Our findings suggest that the answer is no. By stripping away over-engineered workflows and benchmark-specific optimizations, we show that lite agents can be both more faithful and economical, revealing the true capabilities of modern LLMs without hidden scaffolding.

Lita demonstrates that minimal toolkits and lightweight action schemas suffice to solve diverse coding benchmarks, while also reducing overhead in token consumption and design effort. Our ablations further illustrate that agent performance degrades gracefully under simplification, establishing a clear baseline for what constitutes a sufficient design.

We believe the philosophy that less is more. The future of agent design should shift away from handcrafted workflows and toward environments that stimulate genuine model competence. Freeing agents from excessive scaffolding not only benefits evaluation by providing fairer, more authentic comparisons, but also pushes forward the development of LLMs themselves.

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

## A  APPENDIX

### A.1  RELATED WORK

#### A.1.1  AGENT DESIGN PHILOSOPHY EVOLUTION

The evolution of AI agent design represents a fundamental paradigm shift from traditional workflow orchestration toward autonomous reasoning systems. SWE-Agent (Yang et al., 2024) pioneered

repository-level understanding and multistep debugging workflows, it introduced fully autonomous development cycles capable of handling entire feature implementations from planning to deployment. AutoGen (Wu et al., 2024) pioneering multi-agent conversational systems enabling complex task decomposition through structured dialogues, while contemporary enterprise solutions evolved toward sophisticated terminal and cloud-based agents including Claude Code (Anthropic, 2025b)'s terminal-based collaborative architecture for autonomous codebase operations with continuous developer oversight, Gemini CLI (Google, 2025a)'s command-line AI assistance with built-in tools and Model Context Protocol integration, and OpenAI's Codex evolution from code completion to autonomous cloud-based software engineering agents powered by GPT-5-Codex (OpenAI, 2025a) for parallel task execution across entire repositories with comprehensive testing and validation capabilities.

The increasing complexity of comprehensive agentic frameworks has prompted a significant trend toward "lightness" agent philosophies that prioritize minimal architectural overhead while maintaining autonomous capabilities. This reflects recognition that operational simplicity often outweighs architectural sophistication in production environments, exemplified by Mini-SWE-Agent (Yang et al., 2024)'s 100-line Python implementation achieving 68% performance on SWE-bench (Jimenez et al., 2024) benchmarks and Alita (Qiu et al., 2025)'s minimal predefinition approach reaching 75.15% pass@1 accuracy on GAIA (Mialon et al., 2023) benchmarks. This trend represents a fundamental shift toward production-oriented pragmatism, demonstrating that sophisticated problem-solving behavior can emerge from minimal architectural complexity through streamlined interaction patterns rather than maximizing capabilities through complex frameworks.

### A.1.2 LARGE LANGUAGE MODELS FOR CODE

The development of code-specialized large language models has achieved remarkable sophistication through leading proprietary and open-source architectures. Among proprietary systems, Claude 4 (Anthropic, 2025a) introduce hybrid reasoning capabilities that seamlessly transition between rapid responses and extended thinking modes. Gemini 2.5 Pro (Google, 2025b) demonstrates advanced multimodal code understanding through Deep Think reasoning mechanisms. GPT-5 (OpenAI, 2025c) represents unified intelligence architecture with dynamic reasoning effort allocation. The open-source ecosystem demonstrates competitive alternatives through sophisticated architectural innovations. DeepSeek V3 (Liu et al., 2024) employs a 671-billion parameter Mixture-of-Experts design activating only 37 billion parameters per token for computational efficiency, while DeepSeek R1 (Guo et al., 2025) introduces reinforcement learning-optimized reasoning for systematic code verification and multi-step logical problem solving. Qwen3 (Yang et al., 2025) establishes repository-level pretraining strategies across over 40 programming languages with enhanced instruction-following capabilities, offering cost-effective deployment options. Kimi K2 (Team et al., 2025) achieves competitive performance on autonomous coding benchmarks, demonstrating significant improvements in task resolution rates and efficient token utilization. These developments collectively establish code generation as a mature domain where both proprietary and open-source models achieve impressive success rates on real-world software engineering tasks.

### A.1.3 BENCHMARKS FOR CODE AND SOFTWARE ENGINEERING

The evaluation landscape for code-generating systems encompasses traditional function-level benchmarks and emerging agentic frameworks, organized by task categories reflecting evolution from isolated coding assessment toward comprehensive software development evaluation. Code Generation benchmarks establish foundational paradigms through HumanEval (Chen et al., 2021b)'s Pass@k functional correctness metrics and MBPP (Austin et al., 2021)'s programming fundamentals, with recent expansions including BigCodeBench (Zhuo et al., 2024b)'s practical software engineering challenges, LiveCodeBench (Jain et al., 2024)'s contamination-resistant continuous updates, and specialized variants like ClassEval (Du et al., 2023) for class-level generation. Code Reasoning evaluation represents a paradigmatic shift toward execution comprehension through CRUXEval (Gu et al., 2024)'s input-output prediction tasks, revealing significant gaps between generation and understanding capabilities where models excelling at traditional benchmarks struggle with reasoning tasks. Tool Use benchmarks evaluate API interaction capabilities via Berkeley Function Calling Leaderboard (Patil et al., 2024)'s multi-language AST-based assessment and $\tau$-bench (Yao et al., 2024)'s dynamic user-agent conversations with domain-specific tools and behavioral consistency metrics. Agentic Software Development assessment measures autonomous problem-solving through

SWE-Bench (Jimenez et al., 2024)'s real GitHub issues requiring codebase understanding, SWT-Bench (Mündler et al., 2024)'s test generation, and Terminal-bench (Team, 2025)'s command-line interactions, reflecting recognition that modern LLM capabilities require evaluation beyond isolated correctness metrics toward multi-turn interaction, strategic tool usage, and sustained problem-solving assessment, though gaps remain in long-term project development and multi-agent collaboration evaluation, suggesting continued evolution toward comprehensive autonomous programming capability assessment.

## A.2 REPRODUCIBILITY STATEMENT

For our submission, we have uploaded the entirety of the source code as a zipped file that has been properly anonymized. We have organized the codebase such that separate directories correspond to different contributions within the main paper (i.e. dataset collection, evaluation, open source model inference, etc.). The source code contains inline documentation that details purpose and usage of different parts of the codebase. These sections fully cover the logic presented in the code and can be helpful for understanding it. Moving forward, as discussed in the ethics statement, we plan to more formally release Lita to the public as an open source repository with thorough details that describes the benchmark, outlines the code, and details its usage. Because of its easily maintainable design, as discussed in the main paper, our hope and belief is that results should be highly reproducible.

## A.3 THE USE OF LARGE LANGUAGE MODELS

Gemini and OpenAI-GPT were utilized to assist with two primary tasks: 1) Code Generation for Figures: LLMs were used to generate or refine code snippets necessary for the creation of various figures and visualizations within the paper. 2) Paper Writing Polishing: LLMs were employed to review, proofread, and polish the English language and clarity of the manuscript.

## A.4 DETAILED HYPERPARAMETER AND RUNTIME SETTINGS

For the SWE-bench Verified benchmark, we ran all Lita agents, limiting them to 100 iterations and configuring them with temperature=0.0 and top_p=1.0.

## A.5 ADDITIONAL FIGURES AND TABLES

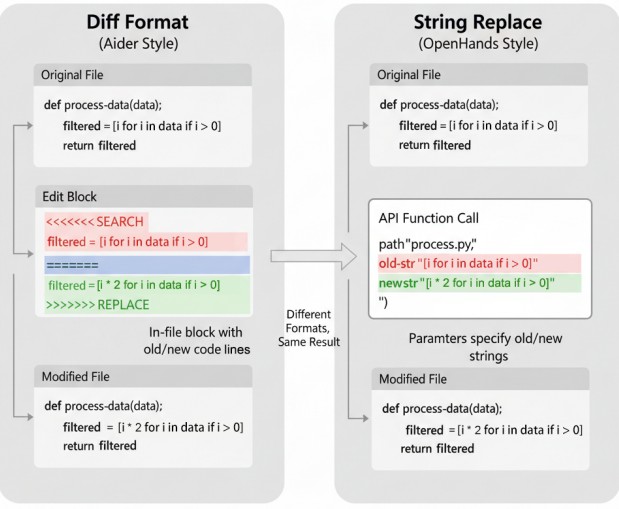

Figure 4: Diff block vs string replace

Table 4: Comparison of pass rates on HumanEval dataset across models with different scaffolds.

| LLM | Scaffold | Pass rate in 30 turns | Pass rate in 1 turn |
|---|---|---|---|
| GPT-4.1 | Lita | 98.2 | 84.8 |
| | OpenHands | 93.9 | 94.5 |
| GPT-4.1-mini | Lita | 97.6 | 94.5 |
| | OpenHands | 97.5 | 97.5 |
| GPT-4o | Lita | 43.6 | 51.2 |
| | OpenHands | - | - |
| GPT-4o-mini | Lita | 16.7 | 14.6 |
| | OpenHands | - | - |
| Claude 3.7 Sonnet | Lita | 100.0 | 95.7 |
| | OpenHands | 100.0 | 97.0 |
| Qwen3-Coder-30B-A3B-Instruct | Lita | 93.9 | 90.9 |
| | OpenHands | - | - |

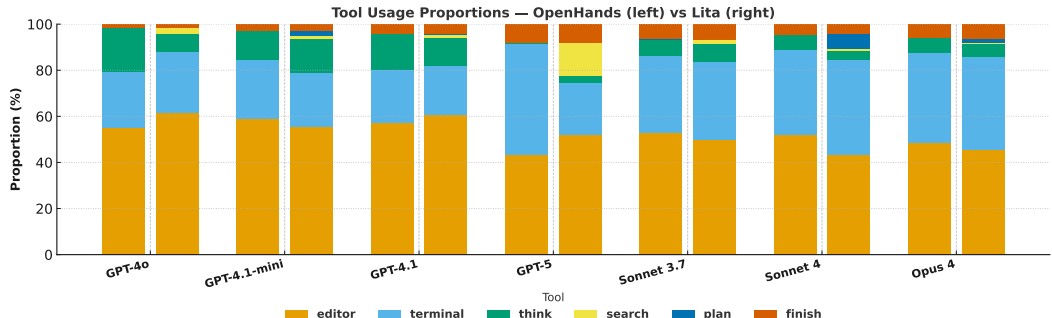

Figure 5: Tool call proportions

## A.6 EXAMPLE PROMPT FOR SWEBENCH

The task prompt prescribed a problem-solving workflow, which constrains the agent's actions and tool use. We believe this also introduces the risk of task data leakage.

```
https://github.com/SWE-agent/mini-swe-agent/blob/main/src/
    minisweagent/config/mini.yaml
```

```
## Recommended Workflow

This workflows should be done step-by-step so that you can iterate
    on your changes and any possible problems.

1. Analyze the codebase by finding and reading relevant files
2. Create a script to reproduce the issue
3. Edit the source code to resolve the issue
4. Verify your fix works by running your script again
5. Test edge cases to ensure your fix is robust
6. Submit your changes and finish your work by issuing the
    following command: `echo COMPLETE_TASK_AND_SUBMIT_FINAL_OUTPUT
    `.
   Do not combine it with any other command. <important>After this
        command, you cannot continue working on this task.</
        important>

## Important Rules
```

1. Every response must contain exactly one action
2. The action must be enclosed in triple backticks
3. Directory or environment variable changes are not persistent. Every action is executed in a new subshell. However, you can prefix any action with `MY_ENV_VAR=MY_VALUE cd /path/to/working/dir && ...` or write/load environment variables from files
....

Table 5: Tool usage distribution across models (OpenHands vs Lita). "Total" shows the total number of tool calls; other rows show the percentage distribution across modes.

| Model | Mode | OpenHands | Lita |
|---|---|---|---|
| **GPT-4.1-mini** | Total | 5838 | 5743 |
| | Editor | 58.9% | 55.4% |
| | Terminal | 25.6% | 23.6% |
| | Think | 12.7% | 14.5% |
| | Search | 0.0% | 1.5% |
| | Plan | 0.0% | 2.1% |
| | Finish | 2.8% | 2.9% |
| **GPT-4.1** | Total | 4652 | 4437 |
| | Editor | 57.2% | 60.7% |
| | Terminal | 22.9% | 21.5% |
| | Think | 15.9% | 11.8% |
| | Search | 0.0% | 1.6% |
| | Plan | 0.0% | 0.2% |
| | Finish | 4.1% | 4.2% |
| **GPT-4o** | Total | 7232 | 7472 |
| | Editor | 55.3% | 61.3% |
| | Terminal | 24.3% | 26.8% |
| | Think | 18.8% | 7.5% |
| | Search | 0.0% | 2.9% |
| | Plan | 0.0% | 0.1% |
| | Finish | 1.6% | 1.4% |
| **GPT-5** | Total | 2416 | 2227 |
| | Editor | 43.1% | 52.0% |
| | Terminal | 48.5% | 22.5% |
| | Think | 0.5% | 3.0% |
| | Search | 0.0% | 14.5% |
| | Plan | 0.0% | 0.0% |
| | Finish | 7.9% | 7.9% |
| **Claude 3.7 Sonnet** | Total | 3204 | 3249 |
| | Editor | 52.9% | 49.8% |
| | Terminal | 33.6% | 33.7% |
| | Think | 7.0% | 8.0% |
| | Search | 0.0% | 1.7% |
| | Plan | 0.0% | 0.1% |
| | Finish | 6.5% | 6.8% |
| **Claude Sonnet 4** | Total | 4829 | 5006 |
| | Editor | 51.8% | 43.2% |
| | Terminal | 37.1% | 41.5% |
| | Think | 6.6% | 3.8% |
| | Search | 0.0% | 0.6% |
| | Plan | 0.0% | 6.5% |
| | Finish | 4.5% | 4.3% |
| **Claude Opus 4** | Total | 3307 | 3465 |
| | Editor | 48.5% | 45.5% |
| | Terminal | 39.1% | 40.4% |
| | Think | 6.7% | 5.5% |
| | Search | 0.0% | 0.4% |
| | Plan | 0.0% | 2.0% |
| | Finish | 5.8% | 6.1% |

Table 6: Results on resolved rate, tokens usage, and cost across models and agents.

| LLM | Agent | Resolved (%) | Tokens (M) | | Cost ($) |
|---|---|---|---|---|---|
| | | | **Input** | **Output** | |
| Claude Opus 4 | Lita | 67.6 | 468.2 | 5.6 | 7441.91 |
| | Lita-mini | 55.2 | 300.4 | 4.7 | 4856.07 |
| Claude Sonnet 4 | Lita | 64.93 | 697.3 | 7.5 | 2204.11 |
| | Lita-mini | 57.8 | 492.3 | 6 | 1566.31 |
| Claude 3.7 Sonnet | Lita | 53.0 | 490.8 | 5.6 | 1556.22 |
| | Lita-mini | 48.6 | 619.7 | 4.3 | 1923.44 |
| GPT-4.1 | Lita | 35.6 | 744.5 | 1.3 | 1499.40 |
| | Lita-mini | 19.6 | 597.2 | 1.1 | 1202.90 |
| GPT-4.1-mini | Lita | 26.4 | 768.3 | 2.7 | 311.69 |
| | Lita-mini | 11.8 | 668.4 | 1.8 | 270.23 |

