# OpenReview forum: "Lita: Light Agent Uncovers the Agentic Coding Capabilities of LLMs"
_ICLR.cc/2026/Conference — Submitted to ICLR 2026_

### Official Review · Reviewer_Qj99 · 2025-10-24

**Soundness:** 3
**Presentation:** 2
**Contribution:** 2
**Rating:** 2
**Confidence:** 4

**Summary:**

This paper addresses issues in current LLM-based code agent design, such as overengineering and unfair evaluation, and proposes a lightweight agent framework named Lita. Its core contributions include:

1. Introducing the concept of a “lightweight agent,” emphasizing minimization of human intervention (e.g., complex workflows, task-specific optimizations) to more accurately assess the coding capabilities of LLMs;
2. Developing the Lita prototype system, which contains only essential tools (e.g., editor, terminal, reasoning module) and supports multi-turn autonomous interaction;
3. Proposing a method to adapt traditional code benchmarks (e.g., HumanEval, SWE-Bench) into an agent evaluation format, ensuring fairness and portability in evaluation;
4. Providing empirical evidence that Lita achieves or surpasses complex baselines (e.g., OpenHands, Aider) on multiple benchmarks, while significantly reducing token consumption and design costs;
5. Proposing the Agent Complexity Law: as model capabilities improve, the performance gap between agents of different complexity will shrink to a negligible level.

**Strengths:**

The paper systematically critiques the issue of “over-scaffolding” in current code agent evaluation and proposes “lightweight design” as a solution. The perspective is novel, and the approach is validated through multi-benchmark (Polyglot, SWE-Bench) and multi-model (GPT, Claude, Qwen series) experiments, demonstrating that the proposed framework offers better cost efficiency compared to other frameworks. It points out that boosting benchmark scores through complex engineering may obscure the model’s true capabilities, and it establishes a practical foundation for a fairer and more sustainable evaluation paradigm, which serves as an important caution for the community.

**Weaknesses:**

- The Agent Complexity Law is derived purely from experimental observations, lacking formal modeling or theoretical derivation (e.g., convergence conditions, mathematical relationships between complexity measures and performance). Moreover, it aligns closely with common industry understanding and intuition — namely, that models with stronger capabilities (e.g., reasoning, instruction-following) are less affected in performance by different frameworks.
- The benchmark coverage remains relatively narrow: it does not include more complex software engineering tasks (e.g., multi-module project debugging, cross-file refactoring) or other mainstream agent scenarios. Furthermore, certain comparative baselines are missing — for instance, on SWE-Bench there is no direct comparison with workflow-based paradigms (such as agentless approaches), which weakens the conclusion that the agent paradigm outperforms workflows.
- The method description is rather general, and the main figure is relatively crude, making it difficult for readers to clearly understand the implementation details of the work.

**Questions:**

- Is there a deeper theoretical explanation for the Agent Complexity Law? Could it be applicable to other, broader domains?
- Although Lita’s toolset (e.g., Editor, Search) is claimed to be “minimal and necessary,” does its selection still rely on subjective judgment? Have the authors considered using automated methods (e.g., analysis of tool usage frequency) to further optimize the toolset?

---

### Official Review · Reviewer_xjWm · 2025-10-31

**Soundness:** 3
**Presentation:** 2
**Contribution:** 3
**Rating:** 6
**Confidence:** 4

**Summary:**

This paper puts forward the idea that current coding agent systems, from more restrictive LLM workflows to more open-ended LLM-based agents are too elaborate and heavily hand-designed for the task or model at hand. This, the authors argue, precludes fair and consistent evaluations between models since various popular scaffolds have been tuned to certain model families and make heavy assumptions about the nature of the task being solved.

The authors then propose a lightweight agent scaffold, with the goal of enabling a more unified evaluation between models with fewer assumptions made. After translating some common coding benchmarks into a consistent agentic form, the authors run their Lita agent (and variants) and demonstrate good pass rates, and token consumption from this simple agent in comparison to more sophisticated scaffolds.

For coding agents, they show that having structured file editing tools are beneficial for weaker agents, and the most parsimonious terminal-only agent does suffice, but only on stronger models.

**Strengths:**

The paper advances a relevant and timely message, which is that on many simple coding and software engineering tasks, the rigid workflows and agent scaffolds that practitioners and researchers alike have used may no longer be acting in our favour as the underlying models are trained to perform these tasks without such heavy-handed assistance. As demonstrated by the results in Table 1, in many cases this can lead to inflated token costs and redundant work.

On the fairness of model evaluations and agentic performance, the paper also makes good observations, for instance that CodeX and OpenHands' prompts are particularly well suited to GPT-series models from OpenAI and moreover focus on SWE-Bench style tasks quite heavily, which is a somewhat contrived subset of software engineering tasks one might undertake with a coding agent.

The paper's "liteness" metric is sound, and the authors do a good job of explaining why the Lita agent is not merely an open-ended loop with a single Bash-based terminal tool.

**Weaknesses:**

One aspect I believe the paper does not dwell on sufficiently is the acknowledgement of the reasons for which a Lita-style agent does well today, and the potential for a continuing need for scaffolds and hand-crafted prompts. Workflows and highly-tuned tool prompts can indeed alter the performance of a model in an agent system, which is highly necessary to get weaker models to do useful work. It may also enable the generation of synthetic data and traces from which to warm-start the next iteration of the model through behaviour cloning or imitation learning. Ostensibly the reason many scaffolds appear over-engineered and heavy-handed today is because the behaviours these were trying to engender have been trained into the strongest base models by commercial model providers. However, the "Lita Design Philosophy" appears to assume that the task at hand closely matches the tasks the underlying model has been extensively trained on. If a model has been trained to write edits using diffs, or to systematically explore a project to gather information before starting a task, or to manage its 'memories', then of course one can (and should) remove prompts and scaffolding intended to induce this behaviour, which are often rigid and brittle. However, if one has a task that the model providers have not yet trained on, or requires more sophisticated and higher-level behaviours, then there is still a case for potentially elaborate prompt-based model behaviour steering and scaffolds.

Another slight weakness of the paper is proposing that the transformation of common coding benchmarks like HumanEval or SWE-Bench Verified is novel or a core contribution. This has been done by practitioners and researchers as part of running agent evals for at least a year now so the novelty of this claim is low. While I agree that having some agreed-upon and widely used evaluation protocol for agent benchmarks would be useful, the descriptiveness of the transformation in Section 2.3 is somewhat terse and lost in the paper. I suggest weakening the claim of this contribution, or substantiating it with far more detail and examples. This could indeed be a separate paper with a more extensive set of transformations on a large number of benchmarks, with an accompanying repository, and clear prescriptions about how to adapt past single-turn benchmarks as well as guidelines for new benchmarks.

**Questions:**

- Why are the Qwen3-Coder solve rates in Table 2 missing for Lita: is this because these models are not able to successfully perform edits or complete the task without support from a more elaborate scaffold?
- The types of 'skills' tested by HumanEval, Aider Polyglot and SWE-Bench Verified are firmly in the distribution of tasks the most powerful commercial models are trained on. Do you believe the advice to build a 'Lite agent' still holds for more difficult tasks that span the range of software engineering tasks (conducting research, systems design, implementing new algorithms, fixing merge conflicts, etc)?
- Do your findings hold outside of coding for LLM agents more broadly?

---

### Official Review · Reviewer_7uPt · 2025-11-01

**Soundness:** 2
**Presentation:** 2
**Contribution:** 2
**Rating:** 2
**Confidence:** 5

**Summary:**

This paper introduces Lita, a lightweight coding agent scaffold that aims to provide a simpler and more extensible framework for analyzing model behavior in coding agent scenarios.

**Strengths:**

1. The motivation is clear: the community currently employs a wide variety of scaffolds, and the lack of disclosure of scaffold details for closed-source models makes fair comparison difficult.
2. The design strives to minimize prompt complexity, avoiding unnecessary sophistication while effectively reducing model cost.

**Weaknesses:**

1. The novelty of Lita’s design is not sufficiently demonstrated. From the perspective of the toolset, Editor and Terminal are standard tools, while Finish and Search are also common (the former appearing as submit in SWE-Agent). Thus, the toolset definition lacks distinctiveness. Similarly, the memory design shows limited originality. The motivation for introducing an additional reasoning module is unclear; if the underlying model is inherently capable of reasoning, this component may be redundant.
2. Several claims appear subjective. For example, in CHALLENGE 1: Fairness, the statement that “OpenHands prompts are particularly well suited to GPT-series models, creating hidden advantages” lacks sufficient evidence, as claude-sonnet4 also performs remarkably well under the OpenHands scaffold.
3. The experiments do not include a comparison with SWE-Agent. Moreover, Mini-SWE-Agent already represents a simpler scaffold and performs comparably to Lita in Table 2, raising the question of whether developing yet another scaffold is necessary.

**Questions:**

please refer to weakness

---

### Meta-Review · Area_Chair_ZWJ1 · 2026-01-07

**Summary:**

Reviewers raised related concerns from different angles.

**Reviewer 7uPt** found the proposed Lita agent to have limited novelty relative to existing lightweight baselines (e.g., Mini-SWE-Agent), questioned the distinctiveness of the toolset and memory design, and noted that several fairness claims were subjective and weakly supported.

**Reviewer Qj99** similarly viewed the Agent Complexity Law as largely intuitive and empirically motivated without theoretical grounding, and highlighted narrow benchmark coverage, missing baselines, and insufficient methodological detail.

**Reviewer xjWm** agreed the motivation is timely but questioned the generality of the conclusions: the benefits of lightweight agents appear tied to tasks already well represented in the training distribution of strong models, and may not extend to more challenging software engineering or non-coding agent settings.

Overall, while the motivation is reasonable, the reviewers agree that limited novelty, insufficiently supported claims, and a narrow evaluation prevent acceptance. Also, the authors did not participate in rebuttal and failed to resolve these concerns.

**Reviewer Concerns:**

The authors did not submit a rebuttal. Consequently, all major reviewer concerns remain unaddressed, including questions about novelty, unsupported claims, limited evaluation scope, missing baselines, and insufficient methodological detail.

**Reviewer Scores:**

The authors did not participate the discussion, so all reviewers did not get an opportunity to re-evaluate this paper.

---

### Decision · Program_Chairs · 2026-01-26

Reject